# ReMIX: Regret Minimization for Monotonic Value Function Factorization in Multiagent Reinforcement Learning

## Abstract

Value function factorization methods have become a dominant approach for cooperative multiagent reinforcement learning under a centralized training and decentralized execution paradigm. By factorizing the optimal joint action-value function using a monotonic mixing function of agents' utilities, these algorithms ensure the consistency between joint and local action selections for decentralized decision-making. Nevertheless, the use of monotonic mixing functions also induces representational limitations. Finding the optimal projection of an unrestricted mixing function onto monotonic function classes is still an open problem. To this end, we propose ReMIX, formulating this optimal projection problem for value function factorization as a regret minimization over the projection weights of different state-action values. Such an optimization problem can be relaxed and solved using the Lagrangian multiplier method to obtain the close-form optimal projection weights. By minimizing the resulting policy regret, we can narrow the gap between the optimal and the restricted monotonic mixing functions, thus obtaining an improved monotonic value function factorization. Our experimental results on Predator-Prey and StarCraft Multiagent Challenge environments demonstrate the effectiveness of our method, indicating the better capabilities of handling environments with non-monotonic value functions.

## 1 Introduction

Reinforcement learning has demonstrated great potential in solving challenging real-world problems, from autonomous driving (Cao et al., 2012; Hu et al., 2019) to robotics and planning (Matignon et al., 2012; Levine et al., 2016; Hüttenrauch et al., 2017). In many scenarios, these tasks involve multiple agents within the same environment and thus require multiagent reinforcement learning (MARL) (Vinyals et al., 2019; Jaques et al., 2019; Baker et al., 2019; Wang et al., 2020b) to coordinate agents and learn desired behaviors from their experiences. Due to practical communication constraints and the need to cope with vast joint action space, MARL algorithms often leverage fully decentralized policies but learn them in a centralized fashion with access to additional information during training. Value function factorization methods, e.g., QMIX (Rashid et al., 2018), QPLEX (Wang et al., 2020a), Qatten (Yang et al., 2020), FOP (Zhang et al., 2021), and DOP (Wang et al., 2020c), have been a dominant approach for such centralized training and decentralized execution (CTDE) MARL (Kraemer & Banerjee, 2016). By factorizing the optimal joint action value function using a monotonic mixing function of per-agent utilities, these algorithms ensure the consistency between joint and local action selections for decentralized decision-making. Superior performance has been reported in many MARL tasks, such as the StarCraft Multiagent Challenge (SMAC) (Samvelyan et al., 2019).

It is known that value function factorization can be viewed as an operator (Dugas et al., 2009), which first computes the optimal joint action value functions as targets and then projects them onto the space representable by monotonic function classes. The projected monotonic mixing functions enable efficient maximization yet allow decentralized decision-making. However, it also poses representational limitations. For instance, QMIX leverages a universal approximator for non-linear monotonic mixing functions. It prevents QMIX from efficiently representing joint action value functions where agents' orderings of their action choices depend on each other (Mahajan et al., 2019). Later, the authors in the paper (Rashid et al., 2020)

proposed an improved projection using Weighted QMIX (WQMIX). It assigns higher weights to the values of optimal joint actions than the suboptimal ones, resulting in a better projection that more accurately represents these optimal values. However, WQMIX relies purely on a heuristic design – such as Centrally-Weighted (CW) and Optimistically-Weighted (OW) – where such weight term is a constant. Finding an optimal projection onto the monotonic function class is still an open problem.

To this end, we propose ReMIX, formulating the optimal projection problem for value function factorization as a regret minimization over the projection weights of different state-action values. Specifically, we construct an optimal policy following the optimal joint action-value function and a restricted policy using its projection onto monotonic mixing functions. A policy regret is then defined as the difference between the expected discounted reward of the optimal policy and that of the restricted policy. By minimizing such policy regret through an upper bound, we can narrow the gap between the optimal and restricted policies and thus force the projected monotonic value function to approach the optimal one during learning, leading to an optimal monotonic factorization with minimum regret. We note that while policy regret minimization has been employed to formulate various optimizations in reinforcement learning, such as optimal prioritized experience replay (Liu et al., 2021) and loss function design (Jin et al., 2018), to the best of our knowledge, this is the first proposal for optimizing value function factorization in MARL through policy regret minimization.

We show that the proposed regret minimization can be solved via the Lagrangian method (Bertsekas, 2014) considering an upper bound. By examining a weighted Bellman equation involving monotonic mixing functions and per-agent critics, we leverage the implicit function theorem (Krantz & Parks, 2002) and derive Karush–Kuhn–Tucker (KKT) (Ghojogh et al., 2021) conditions to find the optimal projection weights in closed form. Our results highlight the key principles contributing to optimal monotonic value function factorization. The optimal projection weights can be interpreted to consist of four components: Bellman error, value underestimates, the gradient of the monotonic mixing function, and the on-policiness of available transitions. We note that the first two terms relating to Bellman error and value underestimates are consistent with the weighting heuristics proposed in WQMIX, thus providing a quantitative justification and recovering WQMIX as a special case. More importantly, our analysis reveals that an optimal value function factorization should also depend on the gradient of the monotonic mixing function and the positive impact of more current transitions.

Following the theoretical results, we provide a tractable approximation of the optimal projection weights and propose a MARL algorithm of ReMIX with regret-minimizing monotonic value function factorization. We validate the effectiveness of ReMIX in Predator-Prey (Böhmer et al., 2020) and SMAC. Compared with state-of-the-art factorization-based MARL algorithms (e.g., WQMIX, QPlex, FOP, DOP), ReMIX is shown to better cope with environments with non-monotonic value functions, resulting in improved convergence and superior empirical performance.

The main contributions of our work are as follows:

- We propose a novel method, ReMIX, formulating the optimal value function factorization as a policy regret minimization and solving the weights of the optimal projection in closed form.

- The theoretical results and tractable weight approximations of ReMIX enable cooperative MARL algorithms with improved value function factorization.

- Experiment results of ReMIX in Predator-Prey and SMAC environments demonstrate superior convergence and empirical performance over state-of-the-art factorization-based methods. We further perform ablation studies to demonstrate the contribution of each component in our design.

## 2 Background

### 2.1 Partially Observable Markov Decision Process

We describe a fully cooperative multiagent sequential decision-making task as a decentralized partially observable Markov decision process (Dec-POMDP) (Oliehoek & Amato, 2016) consisting of a tuple $G = \langle S, U, P, R, Z, O, n, \gamma \rangle$, where $s \in S$ describes the global state of the environment. At each time step, each

agent $a \in A \equiv \{1, \ldots, n\}$ selects an action $u^a \in U$, and all selected actions are combined to form a joint action $\mathbf{u} \in \mathbf{U} \equiv U^n$. This process leads to a transition in the environment based on the state transition function $P(s'|s, \mathbf{u}) : S \times \mathbf{U} \times S \to [0, 1]$. All agents share the same reward function $r(s, \mathbf{u}) : S \times \mathbf{U} \to \mathbb{R}$ with a discount factor $\gamma \in [0, 1)$.

In the partially observable environment, the agents' individual observations $z \in Z$ are generated by the observation function $O(s, u) : S \times A \to Z$. Each agent has an action-observation history $\tau_a \in T \equiv (Z \times U)^*$. Conditioning on the history, the policy becomes $\pi^a(u_a|\tau_a) : T \times U \to [0, 1]$. The joint policy $\pi$ has a joint action-value function: $Q^\pi(s_t, \mathbf{u}_t) = \mathbb{E}_{s_{t+1:\infty}, \mathbf{u}_{t+1:\infty}}[R_t|s_t, \mathbf{u}_t]$, where $t$ is the timestep and $R_t = \sum_{i=0}^{\infty} \gamma^i r_{t+i}$ is the discounted return. In this paper we adopt the centralized training and decentralized execution paradigm: the learning algorithm has access to all local action-observation histories $\boldsymbol{\tau}$ and global state $s$ during training while each agent can only access its own action-observation history in execution.

## 2.2 Policy Regret

The object of MARL is to find a joint policy $\pi$ that can maximize the expected return: $\eta(\pi) = \mathbb{E}_\pi[\sum_{i=0}^{\infty} \gamma^i r_{t+i}]$. For a fixed policy, the Markov decision process becomes a Markov reward process, where the discounted stationary state distribution is defined as $d^\pi(s)$. Considering the partially observable scenario of MARL, we replace the state in discounted state distribution with agents' action observation histories[1], i.e., $d^\pi(\boldsymbol{\tau})$. Similarly, the discounted history action distribution is defined as $d^\pi(\boldsymbol{\tau}, \mathbf{u}) = d^\pi(\boldsymbol{\tau})\pi(\mathbf{u}|\boldsymbol{\tau})$. Then, we will have the expected return rewritten as $\eta(\pi) = \frac{1}{1-\gamma}\mathbb{E}_{d^\pi(\boldsymbol{\tau}, \mathbf{u})}[r(s, \mathbf{u})]$.

We assume there exists an optimal joint policy $\pi^*$ such that $\pi^* = \arg\max_\pi \eta(\pi)$. The regret of the joint policy $\pi$ is defined as $\mathrm{regret}(\pi) = \eta(\pi^*) - \eta(\pi)$. The policy regret measures the expected loss when following the current policy $\pi$ instead of optimal policy $\pi^*$. Since $\eta(\pi^*)$ is a constant, minimizing the regret is consistent with maximizing of expected return $\eta(\pi)$. In this paper, we use regret as an alternative optimization objective for finding the optimal projection in MARL, along with multiple constraints, e.g., the Bellman equation and the sum of projection weights. By minimizing the regret, the current policy $\pi_k$ following a monotonic value factorization will approach the optimum $\pi^*$ following an unrestricted value function.

# 3 Related Work

## 3.1 Value Decomposition Approaches

Value decomposition approaches (Guestrin et al., 2002; Castellini et al., 2019) are widely used in value-based MARL. Such methods integrate each agent's local action-value functions through a learnable mixing function to generate global action values. For instance, VDN (Sunehag et al., 2017) and QMIX estimate the optimal joint action-value function $Q^*$ as $Q_{tot}$ with different formations. VDN aims to learn a joint action-value function $Q_{tot}$ of the sum of individual utilities for each agent. QMIX calculates $Q_{tot}$ by combining mentioned utilities via a continuous state-dependent monotonic function, generated by a feed-forward mixing network with non-negative weights. QTRAN (Son et al., 2019) and QPLEX further extend the class of value functions that can be represented. Besides value-based factorization algorithms, some works extend the value decomposition method to policy-based actor-critic algorithms. In VDAC (Su et al., 2021), a factorized actor-critic framework compatible with A2C can obtain a reasonable trade-off between training efficiency and algorithm performance. Recently proposed FOP (Zhang et al., 2021) provides a new way to factorize the optimal joint policy induced by maximum-entropy MARL into individual policies. DOP (Wang et al., 2020c) addresses the issue of centralized-decentralized mismatch and credit assignment in both discrete and continuous action spaces in the multiagent actor-critic framework. In this paper, we recast the problem of projecting an unrestricted value function onto monotonic function classes as a policy regret minimization, whose solution allows us to find the optimal projection weights to obtain an improved value function factorization.

---

[1]Decentralized MARL problems inherently follow POMDPs, where history-based functions and distributions will reflect the impact of partial observability.

### 3.2 Weighting Scheme in WQMIX

QMIX restricts the joint action-value function to be a monotonic mixing of agents' utilities, such that $Q_{tot}(\boldsymbol{\tau}, \mathbf{u}) = f_s(Q^1(\tau_1, u_1), \ldots, Q^n(\tau_n, u_n))$ where $\frac{\partial f_s}{\partial Q^a} \geq 0, \forall a \in A \equiv 1, \ldots, n$, preventing it from projecting non-monotonic joint action representation. WQMIX solved the limitation by introducing the weights into the projection to retrieve the optimal policy. The WQMIX algorithms - OW and CW QMIXs - can place more importance on the better $Q_{tot}$ in minimizing the loss: $\sum_{i=1}^{b} w(\boldsymbol{\tau}, \mathbf{u})(Q_{tot}(\boldsymbol{\tau}, \mathbf{u}; \theta) - \bar{y}_i)^2$, where $\bar{y}_i = r + \gamma \hat{Q}^*(\boldsymbol{\tau}', \arg\max_{\mathbf{u}'} Q_{tot}(\boldsymbol{\tau}', \mathbf{u}'; \theta^-))$ is the fixed target, $\hat{Q}^*$ is the unrestricted joint action-value function, and $w$ is the weighting function[2]. For example, in OW, the $w$ is given by:

$$w(\boldsymbol{\tau}, \mathbf{u}) = \begin{cases} 1 & Q_{tot}(\boldsymbol{\tau}, \mathbf{u}) < \bar{y}_i \\ \alpha & \text{otherwise.} \end{cases} \tag{1}$$

When a transition is overestimated in the OW paradigm, it will be assigned with a constant weight $\alpha \in (0, 1]$. Compared to OW, CW has a similar mechanism but assigns weights to a transition whose joint action $\mathbf{u}$ is not the best. We note that while insightful, these methods are based on heuristic designs of projection weights. Finding optimal projection weights for monotonic value function factorization is still an open problem. In this paper, we reformulate the problem as a policy regret minimization and solve the optimal projection weights in closed form by relaxing the objective and the Lagrangian method.

## 4 Optimal Projection onto Monotonic Value Functions

### 4.1 Problem Formulation as Regret Minimization

Let $Q^*$ be the unrestricted joint action value function and $Q_{tot} = f_s(Q^1(\tau_1, u_1), \ldots, Q^n(\tau_n, u_n))$ be its estimation obtained through a monotonic mixing function $f_s(\cdot)$ of per-agent utilities $Q^a(\tau_i, u_i)$ for $a = 1, \ldots, n$. For simplicity of notations, we use $Q_k$ to denote $Q_{tot}$ at step $k$. Adopting $\mathcal{B}^* Q_{k-1}^*$ as the target with a Bellman operator $\mathcal{B}^*$, we update $Q_k$ in tandem using a weighted Bellman equation: $Q_k = \arg\min_{Q \in \mathcal{Q}} \mathbb{E}_\mu[w_k(\boldsymbol{\tau}, \mathbf{u})(Q - \mathcal{B}^* Q_{k-1}^*)^2(\boldsymbol{\tau}, \mathbf{u})]$, where $w_k(\boldsymbol{\tau}, \mathbf{u})$ are non-negative projection weights for different transitions that need to be optimized. This projects the unrestricted value function onto a monotonic function class $Q \in \mathcal{Q}$.

To formulate the policy regret with respect to this projection, we consider a Boltzmann policy $\pi_k$ following the agent's individual utilities $Q_k^a$ obtained from such monotonic value factorization, i.e., $\pi_k = [\pi_k^1, \ldots, \pi_k^n]^{\mathrm{T}}$ and $\pi_k^a = e^{Q_k^a(\tau_a, u_a)}/[\sum_{\tau_a, u_a'} e^{Q_k^a(\tau_a, u_a')}]$, as well as a similar policy $\pi^*$ following the unrestricted value function $Q^*$ that is defined over joint actions in the Boltzmann manner. Our objective is to minimize the policy regret $\eta(\pi^*) - \eta(\pi)$ over non-negative projection weights under relevant constraints, i.e.,

$$\begin{aligned} \min_{w_k} \quad & \eta(\pi^*) - \eta(\pi_k) \\ \text{s.t.} \quad & Q_k = \arg\min_{Q \in \mathcal{Q}} \mathbb{E}_\mu[w_k(\boldsymbol{\tau}, \mathbf{u})(Q - \mathcal{B}^* Q_{k-1}^*)^2(\boldsymbol{\tau}, \mathbf{u})], \\ & \mathbb{E}_\mu[w_k(\boldsymbol{\tau}, \mathbf{u})] = 1, \quad w_k(\boldsymbol{\tau}, \mathbf{u}) \geq 0, \\ & Q_k(\boldsymbol{\tau}, \mathbf{u}) = f_s(Q^1(\tau_1, u_1), \ldots, Q^n(\tau_n, u_n)), \end{aligned} \tag{2}$$

where $\pi^*$ and $\pi_k$ are policies in the Boltzmann fashion following the unrestricted and monotonic value functions, respectively. The projection weights must sum up to 1, and $\mu$ is the data distribution that we sample data from the replay buffer. An additional table to summarize and explain the all given notations is provided in Appendix A.

### 4.2 Solving Optimal Projection Weights

The solution to this optimization problem relies on the monotonic function $f_s(\cdot)$ represented by a mixing network, which takes the state and agent networks' output $Q_k^a$ as inputs and generates an estimate of

---

[2]WQMIX defines the weight as $w(s, \mathbf{u})$. Considering Dec-POMDP with the CTDE paradigm, $w(s, \mathbf{u})$ is equivalent to $w(\boldsymbol{\tau}, \mathbf{u})$.

joint value function $Q_{tot}$. Solving the regret minimization problem through the Lagrangian method requires analyzing the KKT conditions. Thus, we first find the first-order derivative of the monotonic mixing network, which will also be leveraged to find an optimal solution. The mixing network is a universal approximator consisting of a two-layer network of non-negative weight (Dugas et al., 2009). We compute its first-order derivative in the following lemma.

**Lemma 1.** *Considering a two-layer mixing network of the weight matrix $W_1, W_2$, bias $b_1, b_2$ and activation function $h(\cdot)$, the derivative of $Q_{tot}$ over one of the local utilities $Q^a$ is:*

$$f'_{s,Q^a} = \frac{\partial Q_{tot}}{\partial Q^a} = h'_{Q^a}(\vec{Q}^{\mathrm{T}} W_1 + b_1) \sum_{j=1}^{m} w^1_{aj} w^2_j,$$

*where $\vec{Q} = [Q^1, \ldots, Q^n]^{\mathrm{T}}$. $W_1, W_2$ are the $n \times m$ and $1 \times m$ matrix correspondingly, with the respective elements $w^1_{ij}$ and $w^2_j$ in each matrix. $n$ is the agent number, and $m$ is the width of the mixing network.*

*Proof.* See Appendix B. $\qquad\square$

Given that the monotonic mixing function is smooth and differentiable, we consider an upper bound of the regret objective (obtained using a relaxation and Jensen's inequality) and formulate its Lagrangian by introducing Lagrangian multipliers with respect to the constraints. It allows us to solve the proposed regret-minimization problem and obtain optimal projection weights in closed form (albeit with a normalization factor $Z^*$).

**Theorem 1** (Optimal weighting scheme). *Under mild conditions, the optimal weight $w_k(s, \mathbf{u})$ to a relaxation of the regret minimization problem in equation 2 with discrete action space is given by:*

$$w_k(\boldsymbol{\tau}, \mathbf{u}) = \frac{1}{Z^*}(E_k(\boldsymbol{\tau}, \mathbf{u}) + \epsilon_k(\boldsymbol{\tau}, \mathbf{u})), \tag{3}$$

*where when $Q_k \leq \mathcal{B}^* Q^*_{k-1}$, we have*

$$E_k(\boldsymbol{\tau}, \mathbf{u}) = \frac{d^{\pi_k}(\boldsymbol{\tau}, \mathbf{u})}{\mu(\boldsymbol{\tau}, \mathbf{u})}(\mathcal{B}^* Q^*_{k-1} - Q_k) \exp(Q^*_{k-1} - Q_k)\left(\sum_{j=1}^{n} \frac{1 - \pi^j}{f'_{s,Q^j}} - 1\right),$$

*and otherwise (i.e., when $Q_k > \mathcal{B}^* Q^*_{k-1}$), we have*

$$E_k(\boldsymbol{\tau}, \mathbf{u}) = 0,$$

*where $Z^*$ is the normalization factor, and $\epsilon_k(\boldsymbol{\tau}, \mathbf{u}))$ is a negligible term when the probability of reversing back to the visited state is small, or the number of steps agents take to revisit a previous state is large.*

*Proof.* We give a sketch of the proof below and provide the complete proof in Appendix C. The derivation of optimal weights consists of the following major steps: (i) Use a relaxation and Jensen's inequality to obtain a more tractable upper bound of the regret objective for minimization. (ii) Formulate the Lagrangian for the new optimization problem and analyze its KKT conditions. (iii) Compute various terms in the KKT condition and, in particular, analyze the gradient of $Q_k$ with respect to weights $p_k$ (defined through the weighted Bellman equation) by leveraging the implicit function theorem (IFT). (iv) Derive the optimal projection weights in closed form by setting the Lagrangian gradient to zero and applying KKT and its slackness conditions.

*Step 1: Relaxing the objective and adopting Jensen's inequality.* To begin with, we replace the original optimization objective function, the policy regret, with a relaxed upper bound. This replacement can be achieved through the following inequality since both sides of the equation have the same minimum:

$$\eta(\pi^*) - \eta(\pi_k) \leq \mathbb{E}_{d^{\pi_k}(\boldsymbol{\tau})}[(Q^*_{k-1} - Q_k)(\boldsymbol{\tau}, \mathbf{u}^*)] + \mathbb{E}_{d^{\pi_k}(\boldsymbol{\tau}, \mathbf{u})}[(Q_k - Q^*_{k-1})(\boldsymbol{\tau}, \mathbf{u})]. \tag{4}$$

The proof of this result is given in Appendix. The key idea is to rewrite the regret using the expectation of the action-value functions with respect to discounted distribution $d^{\pi_k}$. After that, we adopt Jensen's inequality (McShane, 1937) to continue relaxing the intermediate objective function based on a convex function $g(x) = \exp(-x)$. Thus, a new optimization objective generated from equation 4 becomes:

$$\min_{w_k} \quad -\log \mathbb{E}_{d^{\pi_k}(\boldsymbol{\tau})}[\exp(Q_k - Q_{k-1}^*)(\boldsymbol{\tau}, \mathbf{u}^*)] - \log \mathbb{E}_{d^{\pi_k}(\boldsymbol{\tau}, \mathbf{u})}[\exp(Q_{k-1}^* - Q_k)(\boldsymbol{\tau}, \mathbf{u})], \tag{5}$$

where the constraints still hold for the new optimization objective.

*Step 2: Computing the Lagrangian.* In this step, we leverage the Lagrangian multiplier method to solve the new optimization problem in equation 5. For simplicity, we use $p_k$ that absorbs the data distribution $\mu$ into $w_k$. The constructed Lagrangian is:

$$\begin{aligned}
\mathcal{L}(p_k; \lambda, \nu) = & -\log \mathbb{E}_{d^{\pi_k}(\boldsymbol{\tau})}[\exp(Q_k - Q_{k-1}^*)(\boldsymbol{\tau}, \mathbf{u}^*)] \\
& -\log \mathbb{E}_{d^{\pi_k}(\boldsymbol{\tau}, \mathbf{u})}[\exp(Q_{k-1}^* - Q_k)(\boldsymbol{\tau}, \mathbf{u})] \\
& + \lambda(\sum_{\boldsymbol{\tau}, \mathbf{u}} p_k - 1) - \nu^{\mathrm{T}} p_k,
\end{aligned}$$

where $p_k$ is the weight $w_k$ multiplied by the data distribution $\mu$, and $\lambda, \nu$ are the Lagrange multipliers.

*Step 3: Computing the Gradients Required in the Lagrangian.* According to the first constraint in equation 2, the gradient $\frac{\partial Q_k}{\partial p_k}$ can be computed via IFT given by:

$$\frac{\partial Q_k}{\partial p_k} = -[\mathrm{diag}(p_k)]^{-1}[\mathrm{diag}(Q_k - \mathcal{B}^* Q_{k-1}^*)].$$

We also derive the gradient $\frac{\partial d^{\pi_k}(\boldsymbol{\tau}, \mathbf{u})}{\partial p_k}$ for solving the Lagrangian. The derivation details are given in the Appendix.

*Step 4: Deriving the Optimal Weight.* After having the equation for two gradients and an expression of the Lagrangian, we can compute the optimal $p_k$ via an application of the KKT conditions, which needs to set the partial derivative of the Lagrangian equaling to zero, as $\frac{\partial \mathcal{L}(p_k; \lambda, \nu)}{\partial p_k} = 0$, where the optimal weight $w_k$ can be acquired from the $p_k$.

$\square$

The theoretical results shed light on the key factors determining an optimal projection onto monotonic mixing functions. Specifically, the optimal projection weights consist of four components relating to Bellman error, value underestimation, the gradient of the monotonic mixing function, and the on-policiness of available transitions. We will interpret these four components next and develop a deep MARL algorithm through approximations of the optimal projection weights.

*Bellman error $\mathcal{B}^* Q_{k-1}^* - Q_k$:* $Q_k$ is the estimation of the action-value function after the Bellman update. This term measures the distance between the estimation and the Bellman target. A large difference in this term means higher hindsight Bellman error. Due to the KKT slackness condition, our analysis indicates that the optimal projection weight is zero when $Q_k > \mathcal{B}^* Q_{k-1}^*$ is an overestimate of the target value, and otherwise, a higher weight should be assigned when $Q_k$ is more underestimated.

*Value underestimation $\exp(Q_{k-1}^* - Q_k)$:* If $Q_{tot}$ after the Bellman update at current step $k$ is smaller than optimal $Q_{k-1}^*$, it results in an underestimate. In this case, we will assign a higher weight (always larger than 1) to this transition, which is proportional to the exponential of this underestimation gap. In contrast, when overestimating (with a negative gap), the assigned weight becomes lower and always smaller than 1. This is important because an underestimate of function approximation may lead to a sub-optimal $Q_k$ estimation and thus non-optimal action selections.

*Gradient of the mixing network $\sum_{j=1}^n \frac{1 - \pi^j}{f'_{s, Q^j}} - 1$:* It turns out that the optimal projection weights also depend on the inverse of the gradient of the monotonic mixing function $f_s(\cdot)$, which is a new result. Intuitively,

the optimal projection weights would become higher when the monotonic mixing function is insensitive to underlying per-agent utility values (i.e., having a small, positive gradient). We view this result as a form of normalization with respect to different shapes of monotonic mixing function $f_s(\cdot)$. In practical algorithms, we often use the two-layer mixing network with non-negative weights to approximate the monotonic function $f_s(\cdot)$ to produce $Q_k$. The parameters of the mixing network are updated every step, and the gradient value can be readily computed from these parameters. We have provided an instance regarding calculating the gradient of a two-layer mixing network in Lemma 1. It is worth noting that similar gradients can also be obtained for other value function factorization methods.

*Measurement of on-policy transitions* $\frac{d^{\pi_k}(\boldsymbol{\tau},\mathbf{u})}{\mu(\boldsymbol{\tau},\mathbf{u})}$: The efficient update of the joint action value function can be achieved by focusing on transitions that are more possibly to be visited by the current policy, i.e., with a higher $d^{\pi_k}(\boldsymbol{\tau},\mathbf{u})$. Adding this term can speed up the search for the optimal $Q_k$ close to $Q_{k-1}^*$.

### 4.3 Proposed Algorithm

---
**Algorithm 1** ReMIX
---
1: Initialize step, the parameters of mixing network, agent networks, and hyper-network.
2: Set the learning rate $\alpha$ and replay buffer $\mathcal{D}$
3: let $\theta^- = \theta$
4: **for** step $= 1 : \text{step}_{max}$ **do**
5:    $k = 0, s_0 = $ initial state
6:    **while** $s_k \neq$ terminal and $k <$ episode limit **do**
7:       **for** each agent $a$ **do**
8:          $\tau_k^a = \tau_{k-1}^a \cup (o_k, u_{k-1})$
9:          $u_k^a = \begin{cases} \arg\max_{u_k^a} Q(\tau_k^a, u_k^a) & \text{with probability } 1 - \epsilon \\ \text{randint}(1, |U|) & \text{with probability } \epsilon \end{cases}$
10:      **end for**
11:      Obtain the reward $r_k$ and next state $s_{k+1}$
12:      Store the current trajectory into replay buffer $\mathcal{D} = \mathcal{D} \cup (s_k, \mathbf{u}_k, r_k, s_{k+1})$
13:      $k = k + 1, \text{step} = \text{step} + 1$
14:    **end while**
15:    Collect $b$ samples from the replay buffer $\mathcal{D}$ following uniform distribution $\mu$.
16:    **for** each timestep $k$ in each episode in batch $b$ **do**
17:      Evaluate $Q_k$, $Q^*$ and target values
18:      Obtain the utilities $Q_a$ from agents' local networks, and compute the individual policy $\pi_k^a$
19:      Compute the weight:
$$w_k \propto \begin{cases} (\mathcal{B}^* Q_{k-1}^* - Q_k) \exp(Q_{k-1}^* - Q_k) \left( \sum_{j=1}^n \frac{1-\pi^j}{f'_{s,Q^j}} - 1 \right) & \text{when } Q_k \leq \mathcal{B}^* Q_{k-1}^* \\ \epsilon & \text{when } Q_k > \mathcal{B}^* Q_{k-1}^* \end{cases}$$
20:    **end for**
21:    Minimize the Bellman error for $Q_k$ weighted by $w_k$, update the network parameter $\theta$:
     $\theta = \theta - \alpha(\nabla_\theta \frac{1}{b} \sum_i^b w_k (Q_k - y_i)^2)$.
22:    **if** update-interval steps have passed **then**
23:      $\theta^- = \theta$
24:    **end if**
25: **end for**
---

Our analytical results in Theorem 1 identify four key factors determining the optimal projection weights. Interestingly, the first two terms, relating to Bellman error and value underestimation, recover the heuristic designs in WQMIX. Specifically, when the Bellman error of a particular transition is high, which indicates a wide gap between $Q_k$ and $Q_{k-1}^*$, we may consider assigning a larger weight to this transition. Similarly, value underestimation works as a correction term for incoming transitions: based on the difference of current

$Q_k$ and ideal $Q_{k-1}^*$, it will compensate the underestimated $Q_k$ with larger importance while penalizing overestimated $Q_k$ with a smaller weighting modifier, consistent with OW scheme in equation 1.

Additionally, our analysis identifies two new terms: the gradient of the monotonic mixing function and measurement of on-policy transitions, which are crucial in obtaining an optimal projection onto monotonic value function factorization. As discussed, we interpret the gradient term in optimal weights as a form of normalization – by increasing the weights for transitions, where the monotonic mixing function is less sensitive to the underlying per-agent utility, and decreasing the weights otherwise. The measurement of on-policy transitions in the weighting expression emphasizes the useful information carried by more current, on-policy transitions.

Following these theoretical results, we provide a tractable approximation of the optimal projection weights and propose a MARL algorithm, ReMIX, with regret-minimizing projections onto monotonic value function factorizations. The procedure of ReMIX can be found in Algorithm 1. We consider a new loss function with respect to the optimal projection weights $w_k$ applied to the Bellman equation of $Q_k$ (considering $Q_{tot}$ at step $k$), i.e.,

$$L_{\text{ReMIX}} = \sum_{i=1}^{b} \left[ w_i(\boldsymbol{\tau}, \mathbf{u})(Q_k - y_i)^2(\boldsymbol{\tau}, \mathbf{u}) \right], \tag{6}$$

where $b$ is the batch size, and $y_i = \mathcal{B}^* Q_{k-1}^*$ is a fixed target using an unrestricted joint action-value function that can be approximated using a separate network similar to WQMIX.

To compute the projection weights for Bellman error and value underestimation terms, we again leverage the unrestricted joint action-value function $Q^*$ to compute them quantitatively. We note that the Bellman error term also works as the condition in Theorem 1 for deciding whether the weight should be zero. The gradient of the monotonic mixing network can be directly computed using Lemma 1. Ideally, we would also want to include measurement of on-policy transitions term in the calculation, but it is not readily available since distribution $d^{\pi_k}(\boldsymbol{\tau}, \mathbf{u})$ in the numerator is difficult to acquire. Thus, we take an approach similar to existing work (Kumar et al., 2020) and show that the other terms in the derived optimal weights are enough to provide a good estimate and lead to performance improvements. To account for the unknown normalization factor $Z^*$ and improve the stability of the training process, we map the projection weights to a given range, which is modeled as a hyperparameter of our algorithm. We provide numerical results adjusting it in the experiment section.

## 5 Experiment

In this section, we present our experimental results on Predator-Prey and SMAC and demonstrate the effectiveness of ReMIX by comparing the results with several state-of-the-art MARL baselines. Besides, we visualize the optimal weight pattern in heat maps to show the step-wise weight assignment for each transition. Additionally, we conduct the ablation experiments by disabling each term in Theorem 1, and deliver the sensitivity experiments regarding the normalization factor. More details about the environment and hyper-parameter setting are provided in Appendix D. The code of this work is available on GitHub (see supplementary files during the review period).

### 5.1 Predator-Prey

To start with, we consider a complex partially-observable multi-agent cooperative environment, Predator-Prey, that involves 8 agents in cooperation as predators to catch 8 prey on a 10×10 grid. In this task, a successful capture with the positive reward of 1 must include two or more predator agents surrounding and catching the same prey simultaneously, requiring a high level of cooperation. A failed coordination between agents to capture the prey, which happens when only one predator catches the prey, will receive a negative punishment reward. The greater punishment determines the degree of monotonicity. Algorithms that suffer from relative overgeneralization issues or make poor trade-offs in joint action-value function projection will fail to solve this task.

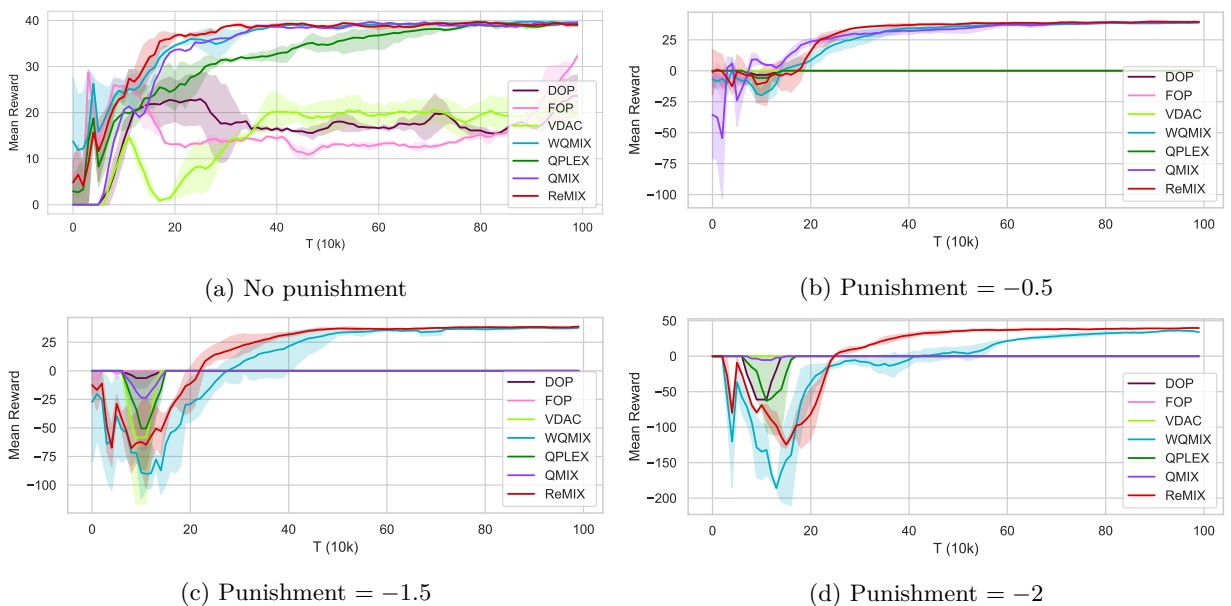

(a) No punishment

(b) Punishment $= -0.5$

(c) Punishment $= -1.5$

(d) Punishment $= -2$

Figure 1: Average reward per episode on the Predator-Prey tasks for ReMIX and other baseline algorithms of 4 settings.

We select multiple state-of-the-art MARL approaches as baseline algorithms for comparison, which include value-based factorization algorithm (i.e., QMIX, WQMIX, and QPLEX), decomposed policy gradient method (i.e., VDAC), and decomposed actor-critic approaches (i.e., FOP and DOP). All mentioned baseline algorithms have shown strength in handling MARL tasks in existing works.

Figure 1 shows the performance of seven algorithms with different punishments, where all results demonstrate the superiority of ReMIX over others. Besides, regarding efficiency, we can spot that ReMIX has the fastest convergence speed in seeking the best policy. In Figure 1c and 1d, ReMIX significantly outperforms other state-of-the-art algorithms in a hard setting requiring a higher level of coordination among agents as learning the best policy with improved joint action representation is required in this setting. Most algorithms, such as QMIX, FOP, and DOP, end up learning a sub-optimal policy where agents learn to work together with limited coordination. Although ReMIX and WQMIX acquired good results eventually, compared to the latter, ReMIX achieves better performance and converges to the optimal policy profoundly faster than WQMIX, demonstrating that our optimal weighting approach can generate a better joint action-value projection.

## 5.2 SMAC

Next, we evaluate ReMIX on the SMAC benchmark. We report the experiments on six maps consisting of one easy map, two hard maps, and three super-hard maps. The selected state-of-the-art baseline algorithms for this experiment are consistent with those in the Predator-Prey environment. The empirical results are provided in Figure 2, demonstrating that ReMIX can effectively generate optimal weight projection for joint actions on SMAC for achieving a higher win rate, especially when the environment becomes substantially complicated and harder, such as *MMM2*. We can see that several state-of-the-art policy-based factorization algorithms are brittle when significant exploration is undergone since joint action representations generated by them are sub-optimal.

Specifically, ReMIX performs well on an easy map *1c3s5z* in Figure 2a, albeit holding the comparable performance among algorithms. On hard maps, such as *3s_vs_5z*, the best policy found by our optimal weighting approach significantly outperforms the remaining baseline algorithms regarding winning rate. For super-hard map *6h_vs_8z*, *MMM2*, and *corridor*, ReMIX, along with QMIX, WQMIX, and QPLEX, can learn a better policy than VDAC, DOP, and FOP. We achieve the highest winning rate by adopting our algorithm

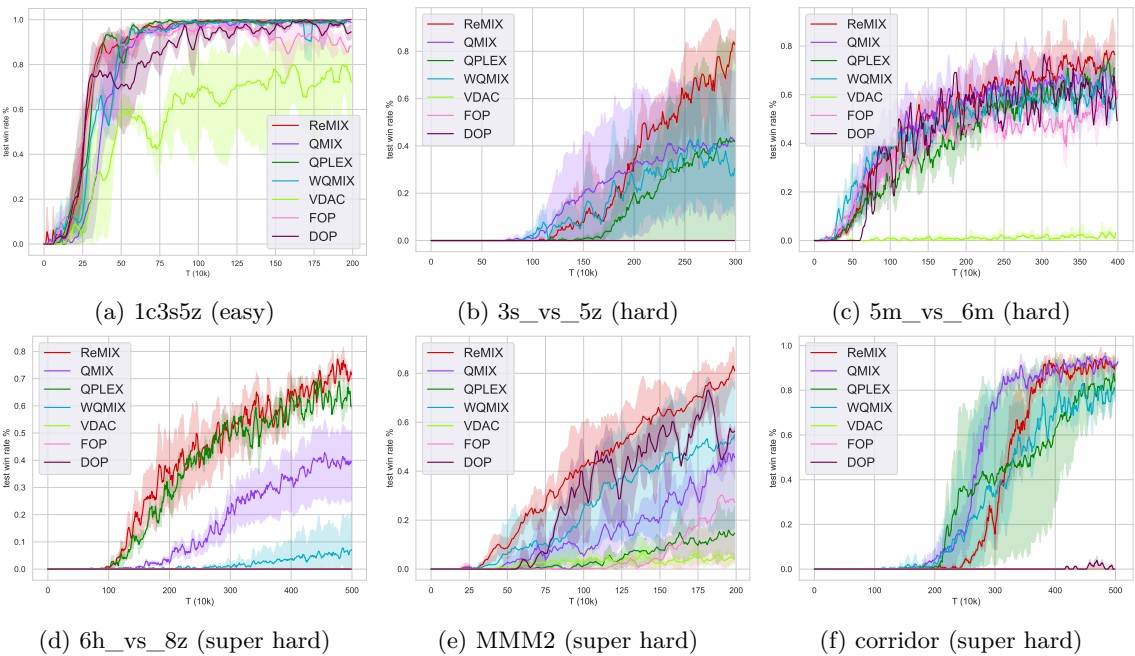

(a) 1c3s5z (easy)  (b) 3s_vs_5z (hard)  (c) 5m_vs_6m (hard)

(d) 6h_vs_8z (super hard)  (e) MMM2 (super hard)  (f) corridor (super hard)

Figure 2: Results of 6 maps (from easy to super hard) on the SMAC benchmark.

on *6h_vs_8z* and *MMM2*. Compared to our method, QMIX and WQMIX suffer from this map as their joint action representations are oblivious to some latent factors, such as the shape of the monotonic mixing network, and therefore fail to generate an accurate joint action representation. On *corridor*, ReMIX manages to learn the model with better performance than WQMIX, QPLEX, and other policy-based algorithms, though standard QMIX has the fastest convergence rate among all baseline algorithms.

### 5.3 Optimal Weight Pattern

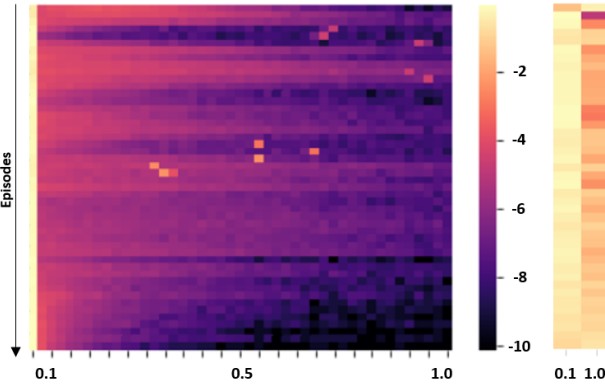

Figure 3: Heatmap pattern of generated optimal weights (left) and WQMIX weights (right) used in the Predator-Prey environment. The training episodes range from 0 to 1M.

In this part, we draw heat maps of the projecting weight probability distributions of ReMIX and WQMIX as the training proceeds to better visualize and compare the weight evolution pattern of transitions sampled as in a minibatch, shown in Figure 3. Adopted weights are generated from the Predator-Prey task with a punishment of -2. We re-scale the absolute value of the transition number to logarithmic probability for scale normalization. As shown in the figure, the probability value of a certain weight is represented by colors, decreasing from 0 in light yellow to -10 in black. The vertical axis represents the training steps, and the

horizontal axis represents the normalized weight value, where ours ranges from 0.1 to 1 and WQMIX is either 0.1 or 1.

The heat map effectively shows the general trend of the weight evolution pattern at different steps. For WQMIX on the Figure 3 right, with the training of the algorithm, the transitions with the smaller weight (0.1) will become more, and those with the larger weight (1) will become fewer. Evolution like this happens since the transitions will approach optimal as the training goes on, while the algorithm will still take all transitions as potential overestimations and assign smaller weights to them as adjustments. A similar evolution pattern can be found in our weight pattern. On the left of Figure 3, during the training, the transitions with higher weights become less, and most transitions will migrate to the bottom right with lower weights, which empirically recovers the heuristic in WQMIX.

Moreover, as an optimal weight projection is used in ReMIX, we will assign different weights to transitions based on evaluating every one of them. We notice that some transitions are assigned with medium weight during the training, given by the light yellow spots on the left of Figure 3. Such a phenomenon demonstrates that the binary-weighted projections in WQMIX are not always accurate. Hence, ReMIX considers all transitions by applying optimal weights to their projections, leading to better results, which also illustrates the performance gap with other algorithms like WQMIX in previous experiments.

### 5.4 Sensitivity Experiment regarding Normalization

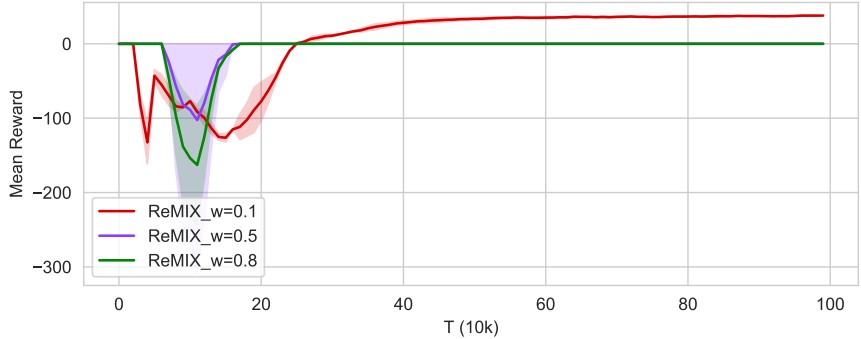

Figure 4: Sensitivity of normalizing the minimum weight to 0.1, 0.5, and 0.8.

We run the experiment in the Predator-Prey environment with a punishment of -1.5 to report the sensitivity with respect to the different normalization of weight ranges. We keep the maximum normalized weight as 1 but test the effects of using different minimums, which are 0.1, 0.5, and 0.8.

As shown in Figure 4, the experiment results are sensitive to the range of the normalized weight. When we map the weight to a minimum of 0.5, the agents in this task can only find a sub-optimal solution. It may be because there exist many overestimations in this task. The joint action representation generated at the is not accurate. Higher minimum weight normalization damages the capability of ReMIX to adjust the projection to retrieve a precise representation rapidly. Therefore, ReMIX performs well under 0.1 to 1 normalization of the weight in this scenario. Note in WQMIX weight is used as $\alpha = 0.1$ for Predator-Prey and $\alpha = 0.5$ for SMAC according to their experiment settings.

### 5.5 Ablation Experiment

For ablations, we conduct experiments by disabling one single term (mentioned in Theorem 1) each at a time to investigate their contribution to finding optimal projection weights, respectively. The ablation results are given in Figure 5. The terms considered in these experiments are Bellman error, value underestimation, and gradient of the mixing network. Figure 4 shows the results on MMM2. Compared to the original result, missing any of the terms will be detrimental to the performance, and the tests without Bellman error have the lowest final winning rate, which is less than 10%. Furthermore, when we turn off the gradient of the

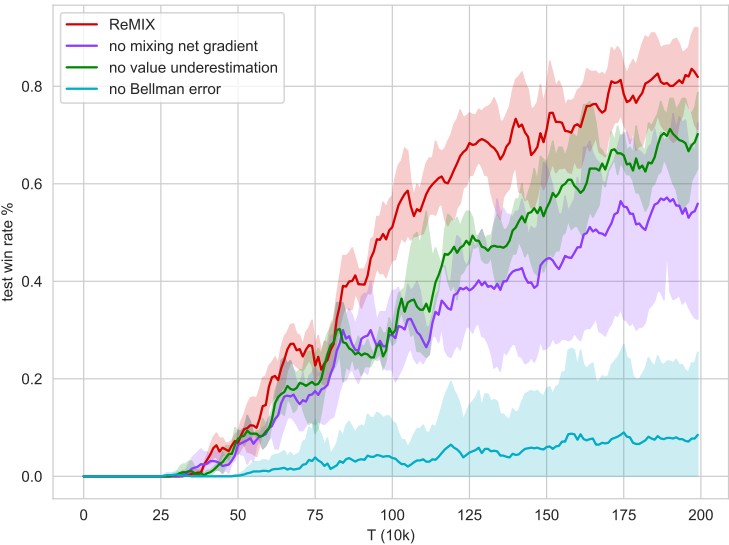

Figure 5: Ablation by disabling one term each for ReMIX on MMM2 (super hard)

mixing network term, the result is only around 60%. Such a phenomenon demonstrates that providing a quantitative weight factorization for the value projection is the critical factor in value-factorization-based MARL tasks. The designing of an optimal weighting scheme without taking the influence of the mixing network into account will be less capable of achieving the ideal final results.

## 6 Conclusion

In this paper, we formulate the optimal value function factorization as a policy regret minimization and solve the optimal projection weights for the cooperative multiagent reinforcement learning problems in closed form. The theoretical results shed light on key factors for an optimal projection. Therefore, we propose ReMIX as a tractable weight approximation approach to enable MARL algorithms with improved value function factorization. Our experiment results in multiple MARL environments show the effectiveness of ReMIX by demonstrating superior convergence and empirical performance over state-of-the-art factorization-based methods.

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

## A  Nomenclature

We use Table 1 to summarize the often-used notations in this paper. More detailed introduction of these notations can be seen in Sections 2, 3, and 4.

Table 1: Definitions of notations.

| Notation | Definition |
|----------|------------|
| $s$ | State of the environment |
| $a$ | Agent |
| $\mathbf{u}$ | Agents' joint action |
| $r$ | Reward |
| $\gamma$ | Discount factor |
| $\boldsymbol{\tau}$ | Joint action-observation history |
| $\pi$ | Joint policy |
| $\pi^*$ | Expected optimal joint policy |
| $Q(\cdot)$ | Action value function |
| $Q_{tot}(\cdot)$ | Monotonic mixing of per-agent action value function |
| $Q^*(\cdot)$ | Unrestricted joint action value function |
| $V(\cdot)$ | Value function |
| $A(\cdot)$ | Advantage function |
| $f_s(\cdot)$ | Monotonic function with input state $s$ |
| $\eta(\pi)$ | Expected return under the joint policy $\pi$ |
| $\mathcal{B}^*$ | Bellman operator, where $\mathcal{B}^* Q(\boldsymbol{\tau}, \mathbf{u}) \overset{\text{def}}{=} r(s, \mathbf{u}) + \gamma \arg\max_{\mathbf{u}'} \mathbb{E}_{\boldsymbol{\tau}'} Q(\boldsymbol{\tau}', \mathbf{u}')$ |
| $w$ | Projection weights of transitions |

## B  Proof of Lemma 1

Considering a two-layer mixing network of the non-negative weight matrix $W_1, W_2$, bias $b_1, b_2$ and activation function $h(\cdot)$. The input $\vec{Q}$ is the vector of all the agents' utilities. Assume there are $n$ agents, $\vec{Q}$ is:

$$\vec{Q} = [Q^1, \ldots, Q^n]^{\mathrm{T}}$$

We assume the mixing network has the width of $m$, based on the input/output dimension, $W_1$ should be a $n \times m$ matrix as:

$$W_1 = \begin{bmatrix} w_{11}^1 & \cdots & w_{1m}^1 \\ \vdots & \ddots & \vdots \\ w_{n1}^1 & \cdots & w_{nm}^1 \end{bmatrix},$$

and $W_2$ is a $m$-dimension vector given by:

$$W_2 = [w_1^2, \ldots, w_m^2]^{\mathrm{T}}.$$

Therefore, $Q_{tot}$ calculated from the utility vector $\vec{Q}$ becomes:

$$f_s(\vec{Q}) = h(\vec{Q}^{\mathrm{T}} W_1 + b_1) W_2^{\mathrm{T}} + b_2. \tag{7}$$

Considering one of the utilities $Q^a$, as long as the derivative of activation $h(\cdot)$ exists ($h(\cdot)$ is smooth and differentiable), based on equation 7, the result is:

$$f'_{s,Q^a} = \frac{\partial Q_{tot}}{\partial Q^a} = h'_{Q^a}(\vec{Q}^{\mathrm{T}} W_1 + b_1) \sum_{j=1}^{m} w_{aj}^1 w_j^2. \tag{8}$$

This concludes the proof.

## C   Proof of Theorem 1

We have provided the outline of the proof including four key steps. In this section, we present the detailed proof of the theorem. The optimization problem needed solving is:

$$\min_{w_k} \quad \eta(\pi^*) - \eta(\pi_k)$$
$$\text{s.t.} \quad Q_k = \arg\min_{Q \in \mathcal{Q}} \mathbb{E}_\mu[w_k(\boldsymbol{\tau}, \mathbf{u})(Q - \mathcal{B}^* Q_{k-1}^*)^2(\boldsymbol{\tau}, \mathbf{u})],$$
$$\mathbb{E}_\mu[w_k(\boldsymbol{\tau}, \mathbf{u})] = 1, \quad w_k(\boldsymbol{\tau}, \mathbf{u}) \geq 0,$$
$$Q_k(\boldsymbol{\tau}, \mathbf{u}) = f_s(Q^1(\tau_1, u_1), \ldots, Q^n(\tau_n, u_n)),$$

This problem is equivalent to:

$$\min_{p_k} \quad \eta(\pi^*) - \eta(\pi_k)$$
$$\text{s.t.} \quad Q_k = \arg\min_{Q \in \mathcal{Q}} \mathbb{E}_{p_k}[(Q - \mathcal{B}^* Q_{k-1}^*)^2(\boldsymbol{\tau}, \mathbf{u})],$$
$$\sum_{\boldsymbol{\tau}, \mathbf{u}} p_k(\boldsymbol{\tau}, \mathbf{u}) = 1, \quad p_k(\boldsymbol{\tau}, \mathbf{u}) \geq 0, \tag{9}$$
$$Q_k(\boldsymbol{\tau}, \mathbf{u}) = f_s(Q^1(\tau_1, u_1), \ldots, Q^n(\tau_n, u_n)),$$

where $p_k = w_k(\boldsymbol{\tau}, \mathbf{u})\mu(\boldsymbol{\tau}, \mathbf{u})$ is the solution to problem equation 9.

To solve the optimization problem in equation 9, we needed to provide some definitions, which are *total variation distance*, *Wasserstein metric*, *the diameter of a set*, and *universal approximator*.

**Definition 1** (Total variation distance). *The total variation distance of the distribution P and Q is defined as $D(P,Q) = \frac{1}{2}\|P - Q\|$.*

**Definition 2** (Wasserstein metric). *For F,G two cumulative distribution functions over the reals, the Wasserstein metric is defined as $d_p(F,G) \overset{\text{def}}{=} \inf_{U,V} \|U - V\|_p$, where the infimum is taken over all pairs of random variables (U,V) with cumulative distributions F and G, respectively.*

**Definition 3** (Diameter of a set). *The diameter of a set A is defined as $\text{diam}(A) = \sup_{x,y \in A} m(x,y)$, where m is the metric on A.*

**Definition 4** (Universal approximator). *A class of function $\hat{\mathcal{F}}$ from $\mathbb{R}^n$ to $\mathbb{R}$ is a universal approximator for a class of functions $\mathcal{F}$ from $\mathbb{R}^n$ to $\mathbb{R}$ if for any $f \in \mathcal{F}$, any compact domain $D \subset \mathbb{R}^n$, and any positive $\epsilon$, one can find a $\hat{f} \in \hat{\mathcal{F}}$ with $\sup_{x \in D} |f(x) - \hat{f}(x)| \leq \epsilon$.*

Though we will leverage trajectories $\boldsymbol{\tau}$ in further derivation, we propose several assumptions using state $s$ for simplicity and consistency with a general definition like the existing practice in (Su & Lu, 2022). The mild assumptions are given as follows:

**Assumption 1.** *The state space S, action space U, and observation space Z are compact metric spaces.*

**Assumption 2.** *The action-value and observation functions are continuous on $S \times U$ and Z, respectively.*

**Assumption 3.** *The transition function T is continuous with respect to $S \times U$ in the sense of Wasserstein metric, which is $\lim_{(s,\mathbf{u}) \to (s_0, \mathbf{u}_0)} d_p(T(\cdot|s, \mathbf{u}), T(\cdot|s_0, \mathbf{u}_0))$.*

**Assumption 4.** *The joint policy $\pi$ is the product of each agent's individual policy $\pi^a(u_a|\tau_a)$.*

**Assumption 5.** *The monotonic mixing function $f_s(\cdot)$ regarding per-agent action-value function $Q^a$ for $\forall a \in A$ is smooth and differentiable.*

These assumptions are not strict and can be satisfied in most MARL environments.

Let $d^{\pi^a}(s)$ denote the discounted state distribution of agent $a$, and $d_i^{\pi^a}(s)$ denote the distribution where the state is visited by the agent for the $i$-th time. Thus, we have:

$$d^{\pi^a}(s) = \sum_{i=1}^{\infty} d_i^{\pi^a}(s), \tag{10}$$

where each $d_i^{\pi^a}(s)$ is given by:

$$d_i^{\pi^a}(s) = (1 - \gamma) \sum_{t_i=0}^{\infty} \gamma^{t_i} \Pr(s_{t_i} = s, s_{t_k} = s, \forall k = 1, ..., i - 1), \tag{11}$$

where the $\Pr(s_{t_i} = s, s_{t_k} = s, \forall k = 1, ..., i - 1)$ in this equation contains the probability of visiting state $s$ for the $i$-th time at $t_i$ and a sequence of times $t_k$, for $k = 1, ..., i$, such that state $s$ is visited at each $t_k$. Thus, state $s$ will be visited for $i$ times at time $t_i$ in total.

The following lemmas are proposed by Liu et al. (2021), where Lemma 2 support the derivation of the Lemma 3, and the latter demonstrates that $\left| \frac{\partial d^{\pi^a}(s)}{\partial \pi^a(s)} \right|$ is a small quantity.

**Lemma 2.** *Let $f$ be an Lebesgue integrable function. $P$ and $Q$ are two probability distributions, $f \leq C$, then:*

$$|\mathbb{E}_{P(x)} f(x) - \mathbb{E}_{Q(x)} f(x)| \leq C \cdot D(P, Q). \tag{12}$$

**Lemma 3.** *Let $\rho$ be the probability of the agent $a$ starting from $(s, u^a)$ and coming back to $s$ at time step $t$ under policy $\pi^a$, i.e. $\Pr(s_0 = s, u_0^a = u^a, s_t = s, s_{1:t-1} \neq s; \pi^a)$, and $\epsilon = \sup_{s, u^a} \sum_{t=1}^{\infty} \gamma^t \rho^{\pi^a}(s, u^a, t)$. We have:*

$$\left| \frac{\partial d^{\pi^a}(s)}{\partial \pi^a(s)} \right| \leq \epsilon d_1^{\pi^a}(s), \tag{13}$$

*where $d_1^{\pi^a}(s) = (1 - \gamma) \sum_{t_1=0}^{\infty} \gamma^{t_1} \Pr(s_{t_1} = s)$ and $\epsilon \leq 1$.*

In the multiagent scenario, each agent only has access to its own trajectory, i.e., the environment is partially observable. Therefore, we replace the state $s$ with agents' observation histories $\boldsymbol{\tau}$ and use the joint action $\mathbf{u}$ with joint policy $\pi$. The conclusions will hold in the mentioned lemmas.

Besides, we have the following additional lemma:

**Lemma 4.** *Given two policy $\pi$ and $\bar{\pi}$, where $\pi = \frac{\exp(Q(\boldsymbol{\tau}, \mathbf{u}))}{\sum_{\mathbf{u}'} \exp(Q(\boldsymbol{\tau}, \mathbf{u}'))}$ is defined by Boltzmann policy, we have:*

$$\mathbb{E}_{\mathbf{u} \sim \bar{\pi}}[Q(\boldsymbol{\tau}, \mathbf{u})] - \mathbb{E}_{\mathbf{u} \sim \pi}[Q(\boldsymbol{\tau}, \mathbf{u})] \leq 1. \tag{14}$$

*Proof.* Suppose there are two joint actions $\mathbf{u}$ and $\bar{\mathbf{u}}$. Let $Q(\boldsymbol{\tau}, \mathbf{u}) = s$, $Q(\boldsymbol{\tau}, \bar{\mathbf{u}}) = t$ and let $s \leq t$.

$$\begin{aligned}
\mathbb{E}_{\mathbf{u} \sim \bar{\pi}}[Q(\boldsymbol{\tau}, \mathbf{u})] - \mathbb{E}_{\mathbf{u} \sim \pi}[Q(\boldsymbol{\tau}, \mathbf{u})] &\leq t - \frac{se^s + te^t}{e^s + e^t} \\
&= t - \frac{s + te^{t-s}}{1 + e^{t-s}} \\
&= t - s - \frac{(t-s)e^{t-s}}{1 + e^{t-s}}.
\end{aligned}$$

Let $f(z) = z - \frac{ze^z}{1+e^z}$, the maximum point $z_0$ satisfies $f'(z) = 0$, from which we further have $1 + e^{z_0} = z_0 e^{z_0}$ where $z_0 \in (1, 2)$. Therefore, we have

$$\mathbb{E}_{\mathbf{u} \sim \bar{\pi}}[Q(\boldsymbol{\tau}, \mathbf{u})] - \mathbb{E}_{\mathbf{u} \sim \pi}[Q(\boldsymbol{\tau}, \mathbf{u})] \leq f(t - s) \leq z_0 - 1 \leq 1.$$

It is worth noting that the derived inequality can also be applied to the situation where we have joint action more than two or we consider the situation regarding per-agent action. $\square$

The following lemma is introduced by Kakade & Langford (2002). It was originally proposed for the finite MDP, while it will also hold for the continuous scenario that is given by Assumption 1 and 2.

**Lemma 5.** *For any policy $\pi$ and $\tilde{\pi}$, we have*

$$\eta(\tilde{\pi}) - \eta(\pi) = \frac{1}{1-\gamma}\mathbb{E}_{d^{\tilde{\pi}}(\boldsymbol{\tau},\mathbf{u})}[A^{\pi}(\boldsymbol{\tau},\mathbf{u})], \tag{15}$$

*where $A^{\pi}(\boldsymbol{\tau},\mathbf{u})$ is the advantage function given by $A^{\pi}(\boldsymbol{\tau},\mathbf{u}) = Q^{\pi}(\boldsymbol{\tau},\mathbf{u}) - V^{\pi}(\boldsymbol{\tau})$.*

**Lemma 6.** *Let $\epsilon_{\pi_k} = \sup_{\boldsymbol{\tau},\mathbf{u}} \sum_{t=1}^{\infty} \gamma^t \rho^{\pi}(\boldsymbol{\tau},\mathbf{u},t)$, the optimal solution $p_k$ to a relaxation of optimization problem in equation 9 satisfies relationship as follows:*

$$p_k(\boldsymbol{\tau},\mathbf{u}) = \frac{1}{Z^*}(D_k(\boldsymbol{\tau},\mathbf{u}) + \epsilon_k(\boldsymbol{\tau},\mathbf{u})), \tag{16}$$

*where when $Q_k \leq \mathcal{B}^* Q_{k-1}^*$, we have $D_k(\boldsymbol{\tau},\mathbf{u}) = d^{\pi_k}(\boldsymbol{\tau},\mathbf{u})(\mathcal{B}^* Q_{k-1}^* - Q_k)\exp(Q_{k-1}^* - Q_k)(\sum_{j=1}^{n} \frac{1-\pi^j}{f'_{s,Q^j}} - 1)$, and when $Q_k > \mathcal{B}^* Q_{k-1}^*$, we have $D_k(\boldsymbol{\tau},\mathbf{u}) = 0$. $Z^*$ is the normalization constant.*

*Proof.* Suppose $\mathbf{u}^* \sim \pi^*$. Let $\pi = \pi^*$ and $\tilde{\pi} = \pi_k$ in Lemma 5, we have

$$
\begin{aligned}
&\eta(\pi^*) - \eta(\pi_k) \\
&= -\frac{1}{1-\gamma}\mathbb{E}_{d^{\pi_k}(\boldsymbol{\tau},\mathbf{u})}[A^{\pi^*}(\boldsymbol{\tau},\mathbf{u})] \\
&= \frac{1}{1-\gamma}\mathbb{E}_{d^{\pi_k}(\boldsymbol{\tau},\mathbf{u})}[V^*(\boldsymbol{\tau}) - Q^*(\boldsymbol{\tau},\mathbf{u})] \\
&= \frac{1}{1-\gamma}\mathbb{E}_{d^{\pi_k}(\boldsymbol{\tau},\mathbf{u})}[V^*(\boldsymbol{\tau}) - Q_k(\boldsymbol{\tau},\mathbf{u}^*) + Q_k(\boldsymbol{\tau},\mathbf{u}^*) - Q_k(\boldsymbol{\tau},\mathbf{u}) + Q_k(\boldsymbol{\tau},\mathbf{u}) - Q^*(\boldsymbol{\tau},\mathbf{u})] \\
&\overset{(a)}{\leq} \frac{1}{1-\gamma}\left[\mathbb{E}_{d^{\pi_k}(\boldsymbol{\tau})}(Q^*(\boldsymbol{\tau},\mathbf{u}^*) - Q_k(\boldsymbol{\tau},\mathbf{u}^*)) + \mathbb{E}_{d^{\pi_k}(\boldsymbol{\tau},\mathbf{u})}(Q_k(\boldsymbol{\tau},\mathbf{u}) - Q^*(\boldsymbol{\tau},\mathbf{u})) + 1\right],
\end{aligned}
\tag{17}
$$

where (a) uses Lemma 4. $\qquad\square$

Since the original optimization is non-tractable, we consider this upper bound to obtain a closed-form solution. Therefore, we replace the objective in equation 9 with the upper bound in equation 17 and solve the relaxed optimization problem, given by

$$
\begin{aligned}
\min_{p_k} \quad & \mathbb{E}_{d^{\pi_k}(\boldsymbol{\tau})}[(Q_{k-1}^* - Q_k)(\boldsymbol{\tau},\mathbf{u}^*)] + \mathbb{E}_{d^{\pi_k}(\boldsymbol{\tau},\mathbf{u})}[(Q_k - Q_{k-1}^*)(\boldsymbol{\tau},\mathbf{u})] \\
\text{s.t.} \quad & Q_k = \operatorname*{arg\,min}_{Q \in \mathcal{Q}} \mathbb{E}_{p_k}[(Q - \mathcal{B}^* Q_{k-1}^*)^2(\boldsymbol{\tau},\mathbf{u})], \\
& \sum_{\boldsymbol{\tau},\mathbf{u}} p_k(\boldsymbol{\tau},\mathbf{u}) = 1, \quad p_k(\boldsymbol{\tau},\mathbf{u}) \geq 0, \\
& Q_k(\boldsymbol{\tau},\mathbf{u}) = f_s(Q^1(\tau_1,u_1),\ldots,Q^n(\tau_n,u_n)),
\end{aligned}
\tag{18}
$$

The derived objective in equation 18 can be further relaxed with Jensen's inequality, given by:

$$\mathbb{E}[g(X)] \geq g(\mathbb{E}[X]), \tag{19}$$

when $g(x)$ is a convex function on real space $\mathbb{R}$.

According to equation 19, we select the convex function $g(x) = \exp(-x)$, and the objective can be further relaxed as:

$$
\begin{aligned}
\min_{p_k} \quad & -\log\mathbb{E}_{d^{\pi_k}(\boldsymbol{\tau})}[\exp(Q_k - Q_{k-1}^*)(\boldsymbol{\tau},\mathbf{u}^*)] - \log\mathbb{E}_{d^{\pi_k}(\boldsymbol{\tau},\mathbf{u})}[\exp(Q_{k-1}^* - Q_k)(\boldsymbol{\tau},\mathbf{u})] \\
\text{s.t.} \quad & Q_k = \operatorname*{arg\,min}_{Q \in \mathcal{Q}} \mathbb{E}_{p_k}[(Q - \mathcal{B}^* Q_{k-1}^*)^2(\boldsymbol{\tau},\mathbf{u})], \\
& \sum_{\boldsymbol{\tau},\mathbf{u}} p_k(\boldsymbol{\tau},\mathbf{u}) = 1, \quad p_k(\boldsymbol{\tau},\mathbf{u}) \geq 0, \\
& Q_k(\boldsymbol{\tau},\mathbf{u}) = f_s(Q^1(\tau_1,u_1),\ldots,Q^n(\tau_n,u_n)),
\end{aligned}
\tag{20}
$$

In order to handle the optimization problem in equation 20, we follow the standard procedures of Lagrangian multiplier method, which is:

$$\mathcal{L}(p_k; \lambda, \nu) = -\log \mathbb{E}_{d^{\pi_k}(\boldsymbol{\tau})}[\exp(Q_k - Q_{k-1}^*)(\boldsymbol{\tau}, \mathbf{u}^*)] - \log \mathbb{E}_{d^{\pi_k}(\boldsymbol{\tau}, \mathbf{u})}[\exp(Q_{k-1}^* - Q_k)(\boldsymbol{\tau}, \mathbf{u})] + \lambda(\sum_{\boldsymbol{\tau}, \mathbf{u}} p_k - 1) - \nu^{\mathrm{T}} p_k,$$

(21)

After constructing the Lagrangian, we further compute some gradients that will be used in calculating the optimal solution. We first calculate the $\frac{\partial Q_k}{\partial p_k}$ according to the implicit function theorem (IFT). Based on the first constraint in equation 20, we aim to find the minimum $Q_k$ to satisfy the $\arg\min(\cdot)$, and therefore we need to ensure the derivative of the term inside $\arg\min(\cdot)$ (we use $f(p_k, Q_k)$ to denote this term) to be zero, which is:

$$f'_{Q_k} = 2 \sum_{\boldsymbol{\tau}, \mathbf{u}} p_k(Q_k - \mathcal{B}^* Q_{k-1}) = 0$$

(22)

We can notice that $F(p_k, Q_k) : f'_{Q_k} = 0$ is an implicit function regarding $Q_k$ and $p_k$. Hence, we apply the IFT on the $F(p_k, Q_k)$ considering the Hessian matrices of $p_k$ and $Q_k$ in $f(p_k, Q_k)$ as follows:

$$\frac{\partial Q_k}{\partial p_k} = -\frac{F'_{p_k}}{F'_{Q_k}} = -\left[\mathrm{diag}(p_k)\right]^{-1} \left[\mathrm{diag}(Q_k - \mathcal{B}^* Q_{k-1}^*)\right].$$

(23)

Next, we derive the expression for $\frac{\partial d^{\pi_k}(\boldsymbol{\tau}, \mathbf{u})}{\partial p_k}$ in the following equation:

$$\begin{aligned}
\frac{\partial d^{\pi_k}(\boldsymbol{\tau}, \mathbf{u})}{\partial p_k} &= \frac{\partial d^{\pi_k}(\boldsymbol{\tau}, \mathbf{u})}{\partial \pi_k} \frac{\partial \pi_k}{\partial Q^a} \frac{\partial Q^a}{\partial Q_k} \frac{\partial Q_k}{\partial p_k} \\
&= \mathrm{diag}(d^{\pi_k}(\boldsymbol{\tau}) + \epsilon_0(\boldsymbol{\tau})) \frac{\partial \pi_k}{\partial Q^a} \frac{\partial Q^a}{\partial Q_k} \frac{\partial Q_k}{\partial p_k} \\
&\overset{(b)}{=} \mathrm{diag}(d^{\pi_k}(\boldsymbol{\tau}) + \epsilon_0(\boldsymbol{\tau})) \mathrm{diag}(\pi_k(1 - \pi_k)) \frac{\partial Q^a}{\partial Q_k} \frac{\partial Q_k}{\partial p_k} \\
&\overset{(c)}{=} d^{\pi_k}(\boldsymbol{\tau}, \mathbf{u})(1 - \pi_k) \frac{1}{f'_{s, Q_k}} \frac{\partial Q_k}{\partial p_k} + \epsilon_0(\boldsymbol{\tau}) \pi_k(1 - \pi_k) \frac{1}{f'_{s, Q_k}} \frac{\partial Q_k}{\partial p_k},
\end{aligned}$$

(24)

where $\epsilon_0(\boldsymbol{\tau}) = \frac{\partial d^{\pi_k}(\boldsymbol{\tau}, \mathbf{u})}{\partial \pi_k(\boldsymbol{\tau})}$ is a small quantity provided by Lemma 3. Besides, (b) is based on the the definition of the Boltzmann policy and Assumption 4, and (c) is based on Assumption 5 the gradient of the monotonic mixing function in Lemma 1.

Since we have all the preparations ready, we now compute the Lagrangian by applying the Karush–Kuhn–Tucker (KKT) condition. We let the Lagrangian gradient to be zero, i.e.,

$$\frac{\partial \mathcal{L}(p_k; \lambda, \nu)}{\partial p_k} = 0$$

(25)

Besides, the partial derivative of the Lagrangian can be computed as:

$$\begin{aligned}
\frac{\partial \mathcal{L}(p_k; \lambda, \nu)}{\partial p_k} &= -\frac{\partial \log \mathbb{E}_{d^{\pi_k}(\boldsymbol{\tau})}[\exp(Q_k - Q_{k-1}^*)(\boldsymbol{\tau}, \mathbf{u}^*)]}{\partial p_k} - \frac{\partial \log \mathbb{E}_{d^{\pi_k}(\boldsymbol{\tau}, \mathbf{u})}[\exp(Q_{k-1}^* - Q_k)(\boldsymbol{\tau}, \mathbf{u})]}{\partial p_k} + \lambda - \nu_{\boldsymbol{\tau}, \mathbf{u}} \\
&= -\frac{1}{Z} \exp(Q_{k-1}^* - Q_k) \left(\frac{\partial d^{\pi_k}(\boldsymbol{\tau}, \mathbf{u})}{\partial p_k} - d^{\pi_k}(\boldsymbol{\tau}, \mathbf{u}) \frac{\partial Q_k}{\partial p_k}\right) + \lambda - \nu_{\boldsymbol{\tau}, \mathbf{u}},
\end{aligned}$$

(26)

where $Z = \mathbb{E}_{\boldsymbol{\tau}', \mathbf{u}' \sim d^{\pi_k}(\boldsymbol{\tau}, \mathbf{u})} \exp(Q^* - Q_k)(\boldsymbol{\tau}', \mathbf{u}')$.

Based on equation 25 and equation 26, and substituting the expression of $\frac{\partial Q_k}{\partial p_k}$ and $\frac{\partial d^{\pi_k}(\tau,a)}{\partial p_k}$ with the derived results in equation 23 and equation 24, we obtain:

$$
\begin{aligned}
p_k(\boldsymbol{\tau},\mathbf{u}) = & \frac{1}{Z(\nu^*_{\boldsymbol{\tau},\mathbf{u}} - \lambda^*)} \left[ d^{\pi_k}(\boldsymbol{\tau},\mathbf{u})(Q_k - \mathcal{B}^* Q^*_{k-1}) \exp(Q^*_{k-1} - Q_k) \left( \sum_{j=1}^{n} \frac{1-\pi^j}{f'_{s,Q^j}} - 1 \right) \right. \\
& \left. + \epsilon_0 \pi_k (Q_k - \mathcal{B}^* Q^*_{k-1}) \exp(Q^*_{k-1} - Q_k) \sum_{j=1}^{n} \frac{1-\pi^j}{f'_{s,Q^j}} \right],
\end{aligned}
\tag{27}
$$

According to Lemma 3, the value of $\epsilon_0$ is smaller than $d^{\pi_k}(\boldsymbol{\tau})$ so the second term will not influence the sign of the equation. Equation 27 will always be larger or equal to zero. By KKT condition, when the $Q_k - \mathcal{B}^* Q^*_{k-1} < 0$, we have $\nu^*_{\boldsymbol{\tau},\mathbf{u}} = 0$. When equation 27 equal to zero, we let $\nu^*_{\boldsymbol{\tau},\mathbf{u}} = 0$ because the value of $\nu^*_{\boldsymbol{\tau},\mathbf{u}}$ will not affect $p_k$. In the contrast, when the $Q_k - \mathcal{B}^* Q^*_{k-1} > 0$, the $p_k$ should equal to zero. Therefore, by introducing a normalization factor $Z^*$, equation 27 can be simplify as follows:

$$
p_k(\boldsymbol{\tau},\mathbf{u}) = \frac{1}{Z^*}(D_k(\boldsymbol{\tau},\mathbf{u}) + \epsilon_k(\boldsymbol{\tau},\mathbf{u})),
\tag{28}
$$

where when $Q_k \leq \mathcal{B}^* Q^*_{k-1}$, we have

$$
\begin{aligned}
D_k(\boldsymbol{\tau},\mathbf{u}) &= d^{\pi_k}(\boldsymbol{\tau},\mathbf{u})(\mathcal{B}^* Q^*_{k-1} - Q_k) \exp(Q^*_{k-1} - Q_k) \left( \sum_{j=1}^{n} \frac{1-\pi^j}{f'_{s,Q^j}} - 1 \right) \\
\epsilon_k &= \epsilon_0 \pi_k (Q_k - \mathcal{B}^* Q^*_{k-1}) \exp(Q^*_{k-1} - Q_k) \sum_{j=1}^{n} \frac{1-\pi^j}{f'_{s,Q^j}}
\end{aligned}
\tag{29}
$$

and when $Q_k > \mathcal{B}^* Q^*_{k-1}$, we have

$$
\begin{aligned}
D_k(\boldsymbol{\tau},\mathbf{u}) &= 0 \\
\epsilon_k &= 0
\end{aligned}
\tag{30}
$$

This concludes the proof.

# D    Environment Details

We use more recent baselines (i.e., FOP and DOP) that are known to outperform QTRAN (Son et al., 2019) and QPLEX (Wang et al., 2020a) in the evaluation. In general, we tend to choose baselines that are more closely related to our work and most recent. This motivated the choice of QMIX (baseline for value-based factorization methods), WQMIX (close to our work that uses weighted projections so better joint actions can be emphasized), VDAC (Su et al., 2021), FOP (Zhang et al., 2021), DOP (Wang et al., 2020c) (SOTA actor-critic based methods). We acquired the results of QMIX, WQMIX based on their hyper-parameter tuned versions from pymarl2(Hu et al., 2021) and implemented our algorithm based on it.

## D.1    Predator-Prey

A partially observable environment on a grid-world predator-prey task is used to model relative overgeneralization problem (Böhmer et al., 2020) where 8 agents have to catch 8 prey in a $10 \times 10$ grid. Each agent can either move in one of the 4 compass directions, remain still, or try to catch any adjacent prey. Impossible actions, i.e., moving into an occupied target position or catching when there is no adjacent prey, are treated as unavailable. If two adjacent agents execute the catch action, a prey is caught and both the prey and the catching agents are removed from the grid. An agent's observation is a $5 \times 5$ sub-grid centered around it, with one channel showing agents and another indicating prey. An episode ends if all agents have been removed or after 200 steps. Capturing a prey is rewarded with r = 10, but unsuccessful attempts by single agents are punished by a negative reward p. In this paper, we consider two sets of experiments with $p = (0, -0.5, -1.5, -2)$. The task is similar to the matrix game proposed by Son et al. (2019) but significantly more complex, both in terms of the optimal policy and in the number of agents.

Table 2: Hyperparameter value settings.

| Hyperparameter | Value |
|---|---|
| Batch size | 128 |
| Replay buffer size | 10000 |
| Target network update interval | Every 200 episodes |
| Learning rate | 0.001 |
| TD-lambda | 0.6 |

## D.2 SMAC

For the experiments on StarCraft II micromanagement, we follow the setup of SMAC (Samvelyan et al., 2019) with open-source implementation including QMIX (Rashid et al., 2018), WQMIX (Rashid et al., 2020), QPLEX (Wang et al., 2020a), FOP (Zhang et al., 2021), DOP (Wang et al., 2020c) and VDAC (Su et al., 2021). We consider combat scenarios where the enemy units are controlled by the StarCraft II built-in AI and the friendly units are controlled by the algorithm-trained agent. The possible options for built-in AI difficulties are Very Easy, Easy, Medium, Hard, Very Hard, and Insane, ranging from 0 to 7. We carry out the experiments with ally units controlled by a learning agent while built-in AI controls the enemy units with difficulty = 7 (Insane). Depending on the specific scenarios(maps), the units of the enemy and friendly can be symmetric or asymmetric. At each time step each agent chooses one action from discrete action space, including noop, move[direction], attack[enemy_id], and stop. Dead units can only choose noop action. Killing an enemy unit will result in a reward of 10 while winning by eliminating all enemy units will result in a reward of 200. The global state information is only available in the centralized critic. Each baseline algorithm is trained with 4 random seeds and evaluated every 10k training steps with 32 testing episodes for main results, and with 3 random seeds for ablation results and additional results.

## D.3 Implementation details and Hyperparameters

In this section, we introduce the implementation details and hyperparameters we used in the experiment. We carried out the experiments on NVIDIA 2080Ti with fixed hyperparameter settings. Recently Hu et al. (2021) demonstrated that MARL algorithms are significantly influenced by code-level optimization and other tricks, e.g. using TD-lambda, Adam optimizer, and grid-searched hyperparameters (where many state-of-the-art are already adopted), and proposed fine-tuned QMIX and WQMIX, which is demonstrated with significant improvements from their original implementation. We implemented our algorithm based on its open-sourced codebase and acquired the results of QMIX and WQMIX from it.

We use one set of hyperparameters for each environment, i.e., no tuned hyperparameters for individual maps. We use epsilon greedy for action selection with annealing from $\epsilon = 0.995$ decreasing to $\epsilon = 0.05$ in 100000 training steps in a linear way. The performance for each algorithm is evaluated for 32 episodes every 1000 training steps. More hyperparameter values are given in Table 2.

