# OpenReview forum: "ReMIX: Regret Minimization for Monotonic Value Function Factorization in Multiagent Reinforcement Learning"
_TMLR — Rejected by TMLR_

### Review · Reviewer_pgFe · 2023-01-12

**Summary Of Contributions:**

This paper tackles the problem of improving value factorization methods in multi-agent reinforcement learning. Given the optimal joint action-value function for multiple agents, previous work has factorized this using a monotonic mixing function of agents' utilities, to ensure consistency between joint and local action selection (we are in the centralized training and decentralized execution). However, this has limitations; this paper endeavours to find the optimal projection of an unrestricted mixing function onto the monotonic function classes. This is done by formulating the problem as regret minimization: the paper compares the optimal policy (following the optimal joint action-value function) to a restricted policy (using its projection onto monotonic mixing functions), with the regret being the difference in discounted reward between the two policies. This is called ReMIX. The paper provides theoretical results, and a tractable approximation that they test in the Predator-Prey and SMAC environments, showing improved performance.

**Audience:**

Yes

**Claims And Evidence:**

Yes

**Requested Changes:**

I think this paper is pretty strong, and would recommend acceptance with very little modification.

Small changes
- Many citations are done in the form: X et al. (2023), instead of (X et al., 2023).
- Note: there are some awkward turns of phrase, e.g.: "Such evolution happens because the transitions will approach optimal with the training going on."  I don't think this significantly impairs the readability of the paper though, so I think it's fine to leave as-is.

**Strengths And Weaknesses:**

Strengths:
+ The paper is pretty clearly written, and the contributions are clear.
+ I have not worked in multi-agent RL for many years. However as far as I can tell, the core idea of the paper, to optimize value function factorization through regret minimization, is novel and interesting. I did not verify the proofs of the theoretical results.
+ The experiments are quite thorough: the paper compares against many different suitable baselines. There are only two environments considered (Predator-Prey and SMAC) but I think these are more than sufficient; SMAC in particular is quite difficult. The paper also includes an ablation experiment disabling different terms of ReMIX in Section 5.5.
+ The results are solid though not outstanding; ReMIX consistently performs among the best algorithms in all environments considered, but is often tied with QMIX or WQMIX (it significantly outperforms in 3/6 SMAC environments, and is mostly similar to WQMIX on the Predator-Prey task)
+ Section 5.3 includes a visualization of the generated optimal weights for ReMIX and WQMIX to better understand why the algorithm works.
+ The authors have committed to making their code open-source
+ The paper includes a discussion of implementation details and hyperparameters in Appendix A.3.3; taken together with the above point, it seems likely that the replicability of the paper is good.

Weaknesses:
- ReMIX is sensitive to hyperparameter choices such as the minimum weight, as seen in Figure 4 (though this isn't a major weakness, and  it is positive that this result is reported)

---

> ### Author Response · Authors · 2023-02-03
> **Responses to reviewer pgFe**
>
> Thank you for your comments and for confirming the contributions of our paper. We provide responses to your questions and concerns as below.
>
> > Reviewer: Many citations are done in the form: X et al. (2023), instead of (X et al., 2023).
>
> ***Response***: We have noticed the format issue in the references cited. We have addressed this problem in the revised version of our paper.
>
> >  Reviewer:  Note: there are some awkward turns of phrase, e.g.: "Such evolution happens because the transitions will approach optimal with the training going on." I don't think this significantly impairs the readability of the paper though, so I think it's fine to leave as-is.
>
> ***Response***:  Thanks for pointing this out. We have performed proofreading carefully and hopefully fixed all typos in the revision.

---

### Review · Reviewer_iDFV · 2023-01-21

**Summary Of Contributions:**

The paper presents a theoretical analysis to justify a closed-form solution for a weighted projection problem that appears in some multi-agent deep reinforcement learning (MARL) methods. This derivation is then used to motivate a MARL algorithm.

**Audience:**

Yes

**Claims And Evidence:**

No

**Requested Changes:**

The rigor of the presentation should be improved. Here are some examples of issues:

- All notations should be explained the first time they are used.
- Some notations should be fixed, e.g., \mu denotes both a distribution and a Lagrange multiplier
- The authors switch back and forth between state-based and history-based distributions, value functions, or policies. I believe the two formulations are not completely equivalent even in the CTDE paradigm. In particular, a discounted stationary state distribution is different from d^\pi(\tau).
- How does the paper from Kumar et al. (2020) show that the measurement of on-policy transitions terms is negligible?
- Should the state/action spaces be compact in Assumption 1? What does smooth mean in Assumption 4? Please specify clearly when those assumptions are used in the proofs.
- Are there any typos in the last equation of page 15? What is d_0^\pi? What is \rho^\pi(s, u, t)?
- The original lemma from Kakade and Langford (2002) applies to finite MDPs. Does it extend to the setting with your current assumptions?
- In the proof of Lemma 6, why is there a white box after (17)? What is d^{\pi_k, \pi^*}, which doesn't seem to appear above?
- In the first sentence below this white box, I don't think that both sides have the same minimum. I guess the authors mean the same argmin. However, even in that case, it's not clear to me that this is true.
- What is \mathbb R_X?

Other:
- References should be between parentheses when they are not part of sentences.
- The paper should be checked for typos.

**Strengths And Weaknesses:**

Strengths
- The presented approach could provide some justification for some previous algorithms.
- The new algorithm seems to work reasonably well in the experiments.

Weaknesses
- Although the paper is generally clear, I find the exposition and writing to be a bit too sloppy, which is problematic for a paper trying to make a theoretical contribution.
- Although the overall theoretical approach seems to be reasonable, it's not completely clear to me that the results hold in the claimed generality.
- The paper contains many typos.

---

> ### Author Response · Authors · 2023-02-03
> **Responses to reviewer iDFV**
>
> We express our deepest appreciation for your time and insightful comments. Here we provide our responses to your questions. Due to the maximum character limit of each reply, we use two separate posts to answer all the questions.
>
> > Reviewer: All notations should be explained the first time they are used.
>
> ***Response***: Thanks for pointing this out. We have added more explanations to them in the latest version. Additionally, we have added a separate table in the appendix to summarize all the notations we need in our paper.
>
> > Reviewer:  Some notations should be fixed, e.g., \mu denotes both a distribution and a Lagrange multiplier
>
> ***Response***:  We adopt another notation $ \nu $  (instead of $\mu $) as the Lagrange multiplier in the revised version to avoid possible misunderstandings.
>
> > Reviewer: The authors switch back and forth between state-based and history-based distributions, value functions, or policies. I believe the two formulations are not completely equivalent even in the CTDE paradigm. In particular, a discounted stationary state distribution is different from d^\pi(\tau).
>
> ***Response***: These two terms are indeed different in nature. While we started with state-based value functions and thus state-based distributions, for POMDPs, these are replaced by history-based value functions and distributions, reflecting the impact of partial observability. We have changed the expression and added a footnote to address this in the revision.
>
>
> > Reviewer:  How does the paper from Kumar et al. (2020) show that the measurement of on-policy transition terms is negligible?
>
> ***Response***: Regarding the on-policy term, ideally, we would also want to include this on-policy term in the calculation. But it is not readily available since we need to compute $ d^{\pi_k}(\boldsymbol{\tau},\mathbf{u}) $ in the numerator. In order to obtain this, we need the transition probability update of every step during the training, which is difficult to acquire. Thus, we take an approach similar to (Kumar et al. 2020) (which focuses on single-agent RL) and show that the other terms in the derived optimal weights are enough to provide a good estimate and lead to performance improvements. Our experiments validate that capturing the three key terms is able to deliver significant improvement. We included some discussions and pointed this out as a direction for future work.
>
> > Reviewer: Should the state/action spaces be compact in Assumption 1? What does smooth mean in Assumption 4? Please specify clearly when those assumptions are used in the proofs.
>
> ***Response***: State/action spaces should be compact. The smoothness of the mixing function means it should be continuously differentiable to apply the first-order KKT conditions. Together they guarantee that limits such as derivatives are properly defined. We have clarified this in the revised paper.
>
>
> > Reviewer:   Are there any typos in the last equation on page 15? What is d_0^\pi? What is \rho^\pi(s, u, t)?
>
> ***Response***: Missing definitions of $ \rho^{\pi}(s,u,t) $: Sorry for the confusion. $ \rho^{\pi}(s,u,t) $ is the probability of an agent starting from $ (s,a) $ and coming back to $ s $ at time $ t $ under the policy $ \pi $, i.e., the probability $ \Pr(s_0=s,a_0=a,s_t=s,s_{1:t-1} \neq s; \pi) $. We have added it. $ d_0 $ is the initial distribution over states. We have also clarified this in the revision.
>
>
> > Reviewer: The original lemma from Kakade and Langford (2002) applies to finite MDPs. Does it extend to the setting with your current assumptions?
>
> ***Response***:  It will suit this setting since most definitions such as $ Q $ and expected return $ \eta $ in both papers are similar. Despite the difference that our paper considers the history-based distribution in alignment with POMDP, we can follow the same process to prove the lemma, and the lemma still holds for continuous MDP. Every $ Q(\boldsymbol{\tau}, \mathbf{u}) $ will satisfy the given lemma, and this is also applicable for value function $ V(\boldsymbol{\tau}) $ and derived advantage function $ A(\boldsymbol{\tau}, \mathbf{u}) $.

---

> > ### Author Response · Authors · 2023-02-03
> > **Responses to reviewer iDFV, continued**
> >
> > >Reviewer: In the proof of Lemma 6, why is there a white box after (17)? What is d^{\pi_k, \pi^*}, which doesn't seem to appear above?
> >
> > ***Response***: The white box at the end of proofs: The white box is created automatically by LATEX, representing the end of the proof. $ d^{\pi_k,\pi*}(\boldsymbol{\tau},\mathbf{u}) $ is a typo and used in an old version of our proof. We removed it.
> >
> > >Reviewer: In the first sentence below this white box, I don't think that both sides have the same minimum. I guess the authors mean the same argmin. However, even in that case, it's not clear to me that this is true.
> >
> > ***Response***: We first relax the original optimization problem in Eq. (4) with an upper bound given by the inequality in Equation (17). This relaxation is tight when $Q_k = Q^*$. Since the original optimization is non-tractable, we consider this upper bound (which yields a convex problem) to obtain a closed-form solution. We will clarify this in the revision. Similar relaxation is also adopted in other papers [1][2] such as using ELBO when computing the mutual information term.
> >
> >
> > > Reviewer: What is \mathbb R_X?
> >
> > ***Response***: This should have been $ \mathbb{R} $, the set of all real values. We have fixed it.
> >
> > > Reviewer: References should be between parentheses when they are not part of sentences.
> >
> > ***Response***: Thanks for pointing them out. We have fixed this format issue in the updated version.
> >
> > > Reviewer: The paper should be checked for typos.
> >
> > ***Response***:  We have performed proofreading carefully and hopefully addressed all typos in the revision.
> >
> >
> > **References**:
> >
> > [1]  Wang, T., Dong, H., Lesser, V., & Zhang, C. (2020). Roma: Multi-agent reinforcement learning with emergent roles.
> >
> > [2]  Christianos, F., Papoudakis, G., Rahman, M. A., & Albrecht, S. V. (2021). Scaling multi-agent reinforcement learning with selective parameter sharing.

---

> > > ### Comment · Reviewer_iDFV · 2023-02-08
> > > **Some issues still remain**
> > >
> > > Thank you for your answers and for updating your manuscript. While some of the points I raised are now addressed, there are still problems in the latest version of the submission regarding the rigor of the presentation. Here are some examples:
> > > - In Section 2.2, \pi^* is introduced as an optimal policy, but later on, it seems it is assumed that it is a Boltzmann policy, e.g., Lemma 6 or below (2), where it seems that the explanation for \pi_k and \pi^*_k are reversed. Also, there's no \pi^*_k in (2)...
> > > - How is the Bellman operator \mathcal B^* defined when everything is expressed in terms of histories?
> > > - The definition of d^\pi_i in the bottom of page 16 still doesn't make sense to me. Take for instance i=2, d^\pi_2 will be defined with index t_2, but inside Pr, t_1 is used and not defined.
> > > - I believe that (10) and (11) are well-defined only in single-agent settings, otherwise you would need an expectation over the actions of the other agents.
> > > - When you switch from state to history, I think you would need to add further assumptions about the observation space and the observation function.
> > >
> > > There are still many typos. For instance:
> > > - d^\pi(\tau) and r(\tau, u) should be defined.
> > > - In the first line of Section 4.1, I believe that Q_i should be Q^i
> > > - The reference to the appendix is missing in the last line of Section 4.1.
> > > - In A.1, Section -> Sections
> > > - Wateriness
> > > - In the proof of Lemma 6, a^* is u^*?

---

> > > > ### Author Response · Authors · 2023-02-10
> > > > **Responses to reviewer iDFV**
> > > >
> > > > We are extremely grateful for your careful and strict examination of our proofs and notations in both the main paper and the appendix. These suggestions will definitely make our paper much more concise.
> > > >
> > > > >Reviewer: In Section 2.2, \pi^* is introduced as an optimal policy, but later on, it seems it is assumed that it is a Boltzmann policy, e.g., Lemma 6 or below (2), where it seems that the explanation for \pi_k and \pi^_k are reversed. Also, there's no \pi^_k in (2)...
> > > >
> > > > ***Response***: Thanks for pointing them out. We have already fixed them in the paper. $ \pi^* $ is the optimal policy. In this paper, we define the policies in a Boltzmann manner, i.e., $ \pi^a={e^{Q^a(\tau_a,u_a)}}/[{\sum_{\tau_a,u_a'} e^{Q^a(\tau_a,u_a')}}] $, including the optimal one. Besides, we checked and reversed the order of $ \pi_k $ and $ \pi^* $ in Lemma 6 and below (2).
> > > >
> > > > >Reviewer: How is the Bellman operator \mathcal B^* defined when everything is expressed in terms of histories?
> > > >
> > > > ***Response***: We have added the history-based Bellman operator $ \mathcal{B}^* $ definition in Appendix A.1, Table 1, where we show the Bellman operator applying to an action-value function regarding trajectories $ \boldsymbol{\tau} $ and joint action $ \mathbf{u} $.
> > > >
> > > > >Reviewer: The definition of d^\pi_i in the bottom of page 16 still doesn't make sense to me. Take for instance i=2, d^\pi_2 will be defined with index t_2, but inside Pr, t_1 is used and not defined.
> > > >
> > > > ***Response***: Thanks for pointing this out. We meant to define d^\pi_i(s) as the discounted probability of visiting state s for the i’th time (thus \sum_i d^\pi_i(s) = d^\pi(s), which is the standard discounted visitation probability for state s). The probability in $ \Pr(\cdot) $ in this equation represents the probability of visiting state $ s $ for the i’th time at $ t_i $. What we were trying to say is that there exists a sequence of times $ t_k $, for $ k=1,..., i $, such that state $ s $ is visited at each $ t_k $, thus for exactly $ i $ times at time $ t_i $.
> > > >
> > > > We notice that this notation and definition are indeed quite confusing. We will change it to $ d^{\pi}_i(s)= (1-\gamma)\sum^\infty_{t_i=0}\gamma^{t_i}\Pr(\exists \t_1,...,t_{i-1}, s.t., s_{t_i}=s, s_{t_k}=s \forall k=1,...,i-1) $ and explain it carefully on page 17. In fact, in this paper, we only need the discounted state probability d^\pi(s) and the first-time visitation probability d^\pi_1(s) (in Lemma 3). We will define these separately to avoid further confusion.
> > > >
> > > >
> > > > >Reviewer: I believe that (10) and (11) are well-defined only in single-agent settings, otherwise you would need an expectation over the actions of the other agents.
> > > >
> > > > ***Response***: In a multi-agent setting, $ \mathbf{u} $ represents the joint action of multiple agents and is drawn from the joint policy $ \pi $. Different from the original (10) and (11), the joint action and policy using $ \mathbf{u} $ and $ \pi $ will be used in the multi-agent scenario. As the reviewer expected, the expectation is taken over the joint actions of multi-agents in this step of our analysis.
> > > >
> > > >
> > > > >Reviewer: When you switch from state to history, I think you would need to add further assumptions about the observation space and the observation function.
> > > >
> > > > ***Response***: Thanks for your advice. We have integrated the assumptions about observation space and function into our current assumptions.
> > > >
> > > > >Reviewer: There are some other typos.
> > > >
> > > > ***Response***: Thanks for pointing them out. Mentioned typos should be fixed in the newest update. Hopefully, there should be no more other issues.

---

> > > > > ### Comment · Reviewer_iDFV · 2023-02-14
> > > > > **Issues with Boltzman policies**
> > > > >
> > > > > Thanks for your reply and for updating your manuscript.
> > > > >
> > > > > Regarding Boltzmann policies, are you assuming that there exists an optimal policy that is a Boltzmann policy or are you considering a best policy among Boltzmann policies? Note that the value function of a Boltzmann policy is generally not a fixed point of the Bellman operator.

---

> > > > > > ### Author Response · Authors · 2023-02-16
> > > > > > **Response to reviewer iDFV**
> > > > > >
> > > > > > Thanks for your comment. Here we provide our clarification to this question.
> > > > > >
> > > > > > >Reviewer: Regarding Boltzmann policies, are you assuming that there exists an optimal policy that is a Boltzmann policy or are you considering a best policy among Boltzmann policies? Note that the value function of a Boltzmann policy is generally not a fixed point of the Bellman operator.
> > > > > >
> > > > > > ***Response***: We consider the Boltzmann policy as a softmax approximation of the optimal policy maximizing $ Q_k $. Since regret optimization requires analyzing first-order optimality conditions while an optimal policy may necessarily not be smooth and differentiable, we consider a Boltzmann policy as its approximation to ensure differentiability.

---

### Review · Reviewer_1Ckq · 2023-01-26

**Summary Of Contributions:**

This article extends the field of cooperative multi-agent reinforcement learning by means of an analytical treatment of the value mixing that is typical in the field, and empirical confirmation that the new insights are practically applicable. The analytical treatment relies on framing the problem as a regret minimisation problem, and provides a decomposition of the contributing factors to the optimal projection of the joint value function into the family of monotonic per-agent value functions. In doing this, the authors recover two components already used in Weighted QMIX, and discover  two more components that have not been previously used in the literature. The empirical results show some modest improvements on StarCraft Multi-Agent Challenge and Predator-Prey environments.

**Audience:**

Yes

**Claims And Evidence:**

Yes

**Requested Changes:**

The citations should be done parenthetically throughout (i.e. using `\citep` rather than `\cite` or `\citet`).

Please explain why the ablations only look at 3 factors. I assume the on-policiness was not used because the algorithm ran was on policy? Should be clarified.

**Strengths And Weaknesses:**

# Strengths

The article is well written and easy to follow. For the most part, the arguments flow well, and there are few structural surprises in the text.

The identification of two new terms for the optimal projection problem into monotonic value functions is novel and useful.

The schematics of the proofs for the main theoretical result are useful and well motivated.

The comparison theoretically and empirically versus Weighted QMIX is appreciated and well explored.

# Weaknesses

The paper would benefit from stating more prominently that this is for fully-cooperative settings. The use of "MIX" in the name is a strong hint, but only for people already familiar with the field. For instance, the title in itself makes it sound like this work could apply to more general cases in MARL, which I don't think it would.

The description in the abstract and the main text of the closed-form optimal projection weights should be clarified that this is an analytical result stemming from framing the problem as a regret minimisation. I was wondering while reading if this was the case, but it was only until Section 4 that it became clear that it was, indeed, a theoretical result.

---

> ### Author Response · Authors · 2023-02-03
> **Responses to reviewer 1Ckq**
>
> We thank the reviewer for the constructive and insightful comments and suggestions. We provide our responses to your questions as follows.
>
> > Reviewer: The citations should be done parenthetically throughout (i.e., using `\citep` rather than `\cite` or `\citet`).
>
> ***Response***:  Thanks for the suggestion. We have addressed them in the revised version.
>
> > Reviewer: The paper would benefit from stating more prominently that this is for fully-cooperative settings. The use of "MIX" in the name is a strong hint, but only for people already familiar with the field. For instance, the title in itself makes it sound like this work could apply to more general cases in MARL, which I don't think it would.
>
> ***Response***: We have updated our abstract and introduction to make it clear that it suits only fully-cooperative MARL settings in the updated version.
>
> > Reviewer: Please explain why the ablations only look at 3 factors. I assume the on-policiness was not used because the algorithm ran was on policy? Should be clarified.
>
> ***Response***:  We did the experiment in an off-policy manner. This term indicates that if we can focus more on the on-policy transitions, it may provide us with better results. However, since distribution $ d^{\pi}(\boldsymbol{\tau}, \mathbf{u}) $ is unreachable as it needs the transition probability update of every step during the training, the measurement of the on-policy transitions term is actually hard to be computed quantitatively. On the other hand, the experimental results with the current three terms have validated that capturing the three key terms is able to deliver significant improvement. We included some discussions and pointed this out as a direction for future work.

---

### Decision · Action_Editors · 2023-03-07

**Recommendation:** Reject

**Comment:**

The main criticism of the paper is that of its technical clarity. Many terms were used without proper definition. Assumptions were made without being stated. Terms were overloaded, incorrectly defined, or ambiguous, and there were typos. Several problems remained even following an initial exchange of discussion with the reviewer. These are serious problems for a paper whose contribution requires a subtle understanding of the interactions between policy regret, partial observability, history vs. state based distributions, single-agent and vs. multiagent. This paper will be especially difficult to understand for this TMLR's readership which does not specialize in MARL. Hence, the long-term impact of this contribution would benefit from carefully refining the technical clarity and presentation.

However, due to the value of the contribution, I strongly encourage the authors to resubmit the paper once the issues with presentation have been fixed.



**Audience:**

The audience is appropriate for TMLR.

However, the general audience of machine learning covered by TMLR is rather large. Since multiagent RL is a niche area within the subfield of RL (which is already a subfield of ML), I think it is especially important to be clear about technical definitions and have theoretical work be accessible to the wider audience.

**Claims And Evidence:**

The paper proposes a novel method for optimal projection of weights onto a monotonic value function. This is valuable because it is a principled approach to the problem of how to assign weights under the value function constraints rather than imposing said constraints and using heuristic methods for updating the the projection weights. The experiments are useful, they show the method is also practical, and are done in commonly used domains. Ablation studies are also included.